# Analysis of Gender Differences in HRV of Patients with Myalgic Encephalomyelitis/Chronic Fatigue Syndrome Using Mobile-Health Technology

**DOI:** 10.3390/s21113746

**Published:** 2021-05-28

**Authors:** Lluis Capdevila, Jesús Castro-Marrero, José Alegre, Juan Ramos-Castro, Rosa M Escorihuela

**Affiliations:** 1Departament de Psicologia Bàsica, Sport Research Institute, Universitat Autònoma de Barcelona, 08193 Bellaterra, Spain; 2ME/CFS Unit, Division of Rheumatology, Vall d’Hebron University Hospital Research Institute, Universitat Autonoma de Barcelona, 08035 Barcelona, Spain; jalegre@vhebron.net; 3Biomedical and Electronic Instrumentation Group, Department of Electronic Engineering, Universitat Politècnica de Catalunya, 08034 Barcelona, Spain; juan.jose.ramos@upc.edu; 4Departament de Psiquiatria i Medicina Legal, Institut de Neurociències, Universitat Autònoma de Barcelona, 08193 Bellaterra, Spain; rosamaria.escorihuela@uab.cat

**Keywords:** mHealth, heart rate variability, HRV, ME/CFS, myalgic encephalomyelitis, chronic fatigue syndrome, gender differences

## Abstract

In a previous study using mobile-health technology (mHealth), we reported a robust association between chronic fatigue symptoms and heart rate variability (HRV) in female patients with myalgic encephalomyelitis/chronic fatigue syndrome (ME/CFS). This study explores HRV analysis as an objective, non-invasive and easy-to-apply marker of ME/CFS using mHealth technology, and evaluates differential gender effects on HRV and ME/CFS core symptoms. In our methodology, participants included 77 ME/CFS patients (32 men and 45 women) and 44 age-matched healthy controls (19 men and 25 women), all self-reporting subjective scores for fatigue, sleep quality, anxiety, and depression, and neurovegetative symptoms of autonomic dysfunction. The inter-beat cardiac intervals are continuously monitored/recorded over three 5-min periods, and HRV is analyzed using a custom-made application (iOS) on a mobile device connected via Bluetooth to a wearable cardiac chest band. Male ME/CFS patients show increased scores compared with control men in all symptoms and scores of fatigue, and autonomic dysfunction, as with women in the first study. No differences in any HRV parameter appear between male ME/CFS patients and controls, in contrast to our findings in women. However, we have found negative correlations of ME/CFS symptomatology with cardiac variability (SDNN, RMSSD, pNN50, LF) in men. We have also found a significant relationship between fatigue symptomatology and HRV parameters in ME/CFS patients, but not in healthy control men. Gender effects appear in HF, LF/HF, and HFnu HRV parameters. A MANOVA analysis shows differential gender effects depending on the experimental condition in autonomic dysfunction symptoms and HF and HFnu HRV parameters. A decreased HRV pattern in ME/CFS women compared to ME/CFS men may reflect a sex-related cardiac autonomic dysfunction in ME/CFS illness that could be used as a predictive marker of disease progression. In conclusion, we show that HRV analysis using mHealth technology is an objective, non-invasive tool that can be useful for clinical prediction of fatigue severity, especially in women with ME/CFS.

## 1. Introduction

Myalgic encephalomyelitis/chronic fatigue syndrome (ME/CFS) is a disorder characterized by medically unexplained and persistent fatigue. Symptoms fluctuating in frequency/severity during the natural course of illness include severe, disabling fatigue, as well as musculoskeletal pain, sleep disturbance, orthostatic intolerance, post-exertional malaise, headaches, and impaired concentration, and short-term memory. Often patients experience distress and disability, which may be exacerbated by a lack of understanding from others [1,2].

A variety of studies have been performed to establish objective biomarkers of ME/CFS disease, including molecular imaging and neuro-functional imaging using magnetic resonance imaging (MRI), magnetoencephalogram (MEG), positron emission tomography (PET), or microRNA expression [3,4]. However, since most of these techniques are expensive and unaffordable, other non-invasive procedures capable of associating a biomarker with fatigue and/or accompanying symptoms are being investigated. To address this, heart rate variability analysis (HRV) could become an objective, non-invasive and easy-to-apply marker. Heart rate variability (HRV) is defined as the fluctuations in the time interval between consecutive beats [5]. HRV is a valid and reliable index to assess the autonomic balance between the sympathetic and the parasympathetic system [6]. Indeed, HRV is considered an index of autonomic resilience, since it reflects the ability to recover from exposure to both physical and psychological stressors [7,8].

HRV is an underlying physiological mechanism that may play an important role in describing autonomic dysfunction in ME/CFS research and clinical practice [9,10]. In a meta-analysis, including 64 articles, it was concluded that differences in heart rate and HRV parameters indicate that ME/CFS patients have altered autonomic cardiac regulation compared to healthy controls [11]. HRV analysis has a prognostic value to predict fatigue severity and exacerbation of autonomic symptoms in ME/CFS, and it is widely used to assess the impact of autonomic imbalance in ME/CFS [12,13]. Previous studies have shown altered sympatho-vagal balance (autonomic modulations) in women with ME/CFS highlighting that lower HRV predicts fatigue status in some patients [14].

HRV is a very affordable marker to track using mobile health (mHealth) technology as it is easy to objectively record the inter-beat interval (RR interval) and apply algorithms to calculate and analyze HRV parameters. In general, using any of the capabilities of a smartphone for assessing or monitoring health or lifestyle can be called mHealth [15]. More specifically, the term mHealth has been defined as “the use of mobile computing and communication technologies in health care and public health” [16].

Returning to ME/CFS, most studies are conducted with samples where women predominate, even though men do also present ME/CFS. For example, in a review about malfunctioning of the autonomic nervous system (ANS) in patients with ME/CFS [13], the authors were able to include twenty-seven articles in the range of methodological quality, of which 25 had only women participants, and 2 had a patient group of men and women. The total number of participants included in those 27 studies was 743 ME/CFS patients, of which 74% (550) were women, and only 26% (193) were men. The only study in that review examining gender differences found that healthy women had a different cardiovascular response to upright tilt compared to healthy men and that this could indicate a predisposition of women to postural insufficiency [17]. In the above-mentioned meta-analysis on the evidence of altered cardiac autonomic regulation in ME/CFS [11], the 64 included studies reported on 2286 ME/CFS patients. Of these studies, 14 recruited exclusively female participants, and 50 studies recruited a mixed-gender sample where 79% were female (*n* = 1803). The authors of this review comment that it was not possible to analyze results based on gender, due to the tendency to report all results within a single sample despite differences in participant characteristics. In another study, 824 ME/CFS patients attending two hospital units were evaluated, of which 74% (550) were women, and only 26% (193) were men, but no gender differences were analyzed [18]. Taken together, this data suggests a higher clinical ME/CFS incidence in women than in men.

Instead, there is substantial evidence of gender differences in reference to the functioning of the ANS. Some studies have suggested a greater reactivity of the parasympathetic system in women compared with men, and greater reactivity of the sympathetic nervous system in men compared with women. In concrete, healthy men have been reported to have high indices of sympathetic function, including muscular sympathetic nerve activity [19], neuron number in sympathetic ganglia [20], and the HRV parameter LF/HF [21]. Other studies found that healthy women show a lower LF/HF power ratio than men, suggesting a preponderance of vagal over sympathetic responsiveness [22,23,24]. Additionally, higher LF power in men has been found in several studies, suggesting the preponderance of sympathetic control of cardiac function [25,26,27,28]. Thus, there is growing evidence suggesting gender differences in ANS activity in chronic conditions, but this has not been fully addressed in ME/CFS patients. For example, most previous neuroimaging studies either limited their subject samples to one gender [29] or did not directly assess gender effects in samples that combined males and females with ME/CFS [30].

Thus, there are few studies on ME/CFS in men, and of the studies that include men, none analyze gender differences using HRV analysis in autonomic dysfunctions derived from ME/CFS. In this study, we aim to: (1) Explore HRV analysis as an objective, non-invasive and easy-to-apply marker of ME/CFS in men, using mHealth technology; (2) assess whether HRV parameters are associated with early self-reported symptoms in men with ME/CFS; and (3) evaluate gender-related differences in HRV and self-reported symptoms in patients with ME/CFS.

## 2. Materials and Methods

### 2.1. Participants

A prospective, cross-sectional cohort study of thirty-two males with ME/CFS was recruited from a single outpatient tertiary-referral center (ME/CFS Clinical Unit, Vall d’Hebrón University Hospital, Barcelona, Spain) from March 2015 to March 2016. All patients were potentially eligible if they met the 1994 CDC/Fukuda criteria [31]. Nineteen age- and sex-matched non-fatigued healthy controls were recruited through word-of-mouth from the hospital and the local community. The 51 men (32 ME/CFS patients and 19 healthy controls) specific for this study were compared with 70 women (45 ME/CFS patients and 25 healthy controls) analyzed in a previous study [14], combined into a unique data set. All participants were of Caucasian descent, from the same geographical area, and had a sedentary lifestyle at the time of the study. Exclusion criteria for the study were previous or current diagnosis of autoimmune conditions, multiple sclerosis, psychosis, major depression, cardiovascular, hematological disorders, infectious diseases, sleep apnea or thyroid-related illnesses, pregnancy or breast-feeding, smoking, hormone-related drugs, and symptoms of ME/CFS that did not fit to the case criteria used for this study. All participants voluntarily provided written, signed informed consent prior to study participation.

### 2.2. General Procedure for Data Collection

The general procedure was identical to the one followed in the previous study with female ME/CFS patients and healthy control participants [14]. Briefly, all eligible participants attended the first interview where sociodemographic and self-reported outcome questionnaires were collected. They were asked about their fatigue severity, neurovegetative complaints, sleep quality, anxiety and depression, and symptoms of autonomic dysfunction. After the first interview, participants were assigned dates for the following additional HRV recording sessions.

### 2.3. Measures

The following symptom assessment tools were used to evaluate all participants under the supervision of two trained investigators who ensured compliance. A validated Spanish translation of the validated English version was used for all questionnaires.

#### 2.3.1. Neurovegetative Complaints Questionnaire (NCQ)

The original Neurovegetative Complaints Questionnaire (NCQ) consisted of 28 items concerning headaches, problems with falling asleep, restlessness, chest pain, indigestion, slowness of working, sensitivity to light, effort, flushing, concentration, dyspnea, preference for being left alone, tiredness, fainting, heart palpitations, noise, difficulty with doing two tasks simultaneously, preference for working at one’s own pace, dizziness, depression, wet hands, crying spells, altered libido, irritability, lack of initiative, awakening at night, defeatism, and not being appreciated by others. Participants had to indicate the frequency of occurrence of these symptoms on a 4-point Likert scale (1 = no, never; 2 = yes, sometimes; 3 = yes, regularly, and 4 = yes, often). Higher scores indicated more autonomic complaints [32].

#### 2.3.2. Fatigue Impact Scale

The Fatigue Impact Scale (FIS-40) is a 40-item questionnaire to assess fatigue symptoms as part of an underlying chronic condition. It includes three domains reflecting the perceived feeling of fatigue: Physical (10-items), cognitive (10-items), and psychosocial functions (20-items). Each item is scored from 0 (no fatigue) to 4 (severe fatigue). The overall score is calculated by adding together the responses to the 40 questions (ranging from 0–160). Higher scores indicate increased functional limitations, due to fatigue [33].

#### 2.3.3. Composite Autonomic Symptom Scale

For measuring autonomic dysfunction, all participants were screened using the Composite Autonomic Symptom Scale (COMPASS-31), a 31-item questionnaire designed to evaluate the frequency and severity of autonomic function symptoms, grouped in six domains: Orthostatic intolerance (4-items), vasomotor (3-items), secretomotor (4-items), gastrointestinal (12-items), bladder (3-items) and pupillomotor systems (5-items). Added together, the six domain scores provide a total COMPASS-31 score ranging from 0 to 100—with higher scores indicating more severe autonomic complaints [34].

#### 2.3.4. Pittsburgh Sleep Quality Index

The Pittsburgh Sleep Quality Index (PSQI) is the most commonly used sleep quality questionnaire in research settings, consisting of 19-item self-administrated questions to assess sleep disturbances over the previous month. Scores are acquired on each of seven components of sleep quality: Subjective sleep quality, sleep latency, sleep duration, habitual sleep efficiency, sleep disturbances, use of sleeping medication, and daytime dysfunction. Each domain is scored from 0 to 3 (0 = no sleep problems and 3 = severe sleep problems). The overall PSQI score ranges from 0 to 21 points, with a score of ≥ 5 indicating poorer sleep quality [35].

#### 2.3.5. Hospital Anxiety and Depression Scale

To assess anxiety and depression symptoms, the Hospital Anxiety and Depression Scale (HADS) was used. The HADS is a validated 14-item self-reported measure (seven associated with anxiety symptoms and 7 with depression) that addresses the preceding week in an outpatient clinical practice. Each item is scored on a 4-point Likert scale (e.g., as much as I always do 0 = not quite so much; 1 = definitely not so much; 2 = and not at all; 3 = giving maximum subscale scores of 21 for depression and anxiety, respectively. Scores of 0–7 are interpreted as normal, 8–10 as mild symptoms, 11–14 as moderate, and 15–21 as severe for either anxiety or depression. The total HADS score ranges from 0 (no anxiety/depression) to 42 (severe anxiety/depression) [36].

#### 2.3.6. Heart Rate Variability Recording and Analysis

The assessment of heart rate variability was performed between 3:00 p.m. and 6:00 p.m. in a semi-dark room at the local ME/CFS Clinical Unit, maintained at a temperature between 20–24 °C. Participant’s session ratings of inter-beat cardiac intervals (RR intervals) were continuously monitored/recorded over three 5-min periods on different days and weeks. These three different recordings were done with a time interval of 7 to 21 days between 2 consecutive recordings. For the final analysis of each variable, the values obtained in the three independent recordings were averaged. Participants were asked to abstain from caffeine, alcohol, and physical activity for 12 h prior to testing. In each session, participants were directed to lie supine without speaking or moving for the duration of the session. After five minutes of rest, the heart rate variability data was registered, recording continuously for five minutes of natural breathing [37].

The RR intervals were detected with a wearable cardiac chest band (Polar Band H7; Polar Electro, Finland). This band sends every second a data packet containing the average heart rate and all the RR intervals detected since the last packet with a resolution of 1/1024 s. The field containing the RR intervals is empty if a heartbeat has not been detected, since the last transmission. The information is sent using a Bluetooth Low Energy (BLE) connection to minimize energy consumption. All the RR intervals received were converted to a resolution of 1 ms and saved locally in a mobile device with the FitLab^®^ App (Health and SportLab, Barcelona, Spain). Figure 1 shows different screenshots of the app with prompts for completing data related to the recording conditions.

The analysis of the RR series was performed on a remote server. The system permitted HRV recordings, while checking the quality of the data in real-time. The accuracy and reliability of the Polar cardiac band were previously tested against the gold standard based on the ECG [38]. For HRV analysis, we followed the recommendations of the Task Force of the European Society of Cardiology and the North American Society of Pacing and Electrophysiology [39]. All the recordings were pre-analyzed to detect artifacts and erroneous data before the HRV analysis. A maximum error of 10% of RR intervals was accepted and filtered. The error correction of the RR series was based on outliers´ detection. The median of the last 10 RR intervals was used. The outliers were classified in false positives, false negatives, or ectopic beats. A correction in the RR series was applied to keep the total duration of the recording [40].

For the time-domain analysis, the mean of RR intervals (meanRR), the standard deviation of all RR intervals (SDNN), the root mean square of differences (RMSSD) of successive RR intervals, and the mean number of times an hour in which the change in successive normal sinus (RR) intervals exceeds 50 msec (pNN50) were calculated. For frequency-domain analysis, all RR series were resampled at 4 Hz using a shape-preserving piecewise cubic interpolation prior to the HRV analysis. The power spectrum of the resampled time series was estimated using the Fast Fourier Transform after removing the mean of the time series and multiplying the time series by a Hann window [37]. The power densities in the very low frequency (VLF) band (0.00–0.04 Hz), the low frequency (LF) band (0.04–0.15 Hz), and the high frequency (HF) band (0.15–0.40 Hz) were calculated from each 5-min spectrum by integrating the spectral power density in the respective frequency bands. Additional calculations included the LF/HF ratio, as well as the normalized LF and HF values (LFnu and HFnu, respectively).

### 2.4. Data Analysis

All calculations were performed with the IBM SPSS Statistics package for Mac OS (version 21). Calculation of HRV parameters was carried out with FitLab software (HealthSportLab.com, Barcelona, Spain) and MATLAB environment (MathWork, Natick, MA, USA). In the first part of Section 3, the differences between the newly reported data from the two groups of men were assessed with the t-test for independent samples, and the Mann-Whitney U non-parametric test when appropriate. Pearson correlation analyses were performed to test bivariate associations between questionnaire scores and HRV indices, or between questionnaire scores and themselves. To evaluate the relative relationship between the most significant HRV indices and fatigue dysfunction, we performed separated linear regression analyses in control subjects and ME/CFS patients. The significance threshold was set at *p* < 0.05. In the second part of Section 3, we report the differential gender effects and interactions with the ME/CFS fatigue condition shown by the two-way analysis of variance (ANOVA) applied to HRV data of male and female participants. HRV variables for the men were obtained from recordings of the participants included in the current study, whereas the data for women correspond to the participants reported in a previous, recently published article [14]. Gender (male and female) and health condition (control, ME/CFS) were used as independent variables in the two-way ANOVA, and Duncan’s test was used for post-hoc comparisons between pairs of groups. HRV data from two participants, one of the ME/CFS women and one of the control men, were discarded, due to unsuccessful processing of HRV recordings.

## 3. Results

### 3.1. Demographic and Clinical Characteristics of Participants (Men)

Table 1 shows descriptive statistics for the control group and ME/CFS patient men. There were no differences in age and BMI between ME/CFS patients and controls. Heart rate and systolic arterial pressure were higher in ME/CFS patients compared with control participants (*p* < 0.05). Diastolic arterial pressure was also high in ME/CFS patients (*p* = 0.054).

### 3.2. Self-Reported Measures (Men)

The neurovegetative complaints (NCQ), as well as the physical, cognitive, psychosocial, and total fatigue (FIS-40) scores reported by ME/CFS patients, were much higher than those reported by healthy controls, thus being a robust between-group significant difference (*p* < 0.001; Table 1). The higher scores on those measures are consistent with the diagnosis of ME/CFS. In the COMPASS-31 questionnaire, the ME/CFS patients reported increased orthostatic intolerance, vasomotor, secretomotor, bladder, pupillomotor (all *p* < 0.001), gastrointestinal (*p* = 0.001) and total COMPASS-31 score (*p* < 0.001), compared with the control participants (Table 1). These results are also consistent with the increased number of autonomic symptoms reported in the neurovegetative complaints questionnaire (NCQ; Table 1). In the PSQI questionnaire, the ME/CFS patients had increased symptoms referring to subjective sleep quality, sleep latency, sleep efficiency (all *p* < 0.001), and sleep duration (*p* = 0.001), compared with the control participants. They also reported increased sleep disturbances, greater use of sleep medication, and increased daytime dysfunction compared with control participants (*p*< 0.001; Table 1). The total PSQI score was three and a half times higher in ME/CFS patients compared with controls (*p* < 0.001; Table 1). ME/CFS patients reported higher scores (10–11 points more) of anxiety and depression in the HADS questionnaire, compared with controls (*p* < 0.001; Table 1).

### 3.3. Heart Rate Variability Indices (Men)

All recordings were analyzed for the presence of artifacts. Recordings with more than 10% of artifacts were rejected (6% of the total); 75% of the recordings were artifact-free, and the remaining were corrected. The signal error average for all RR records of the study was 0.72%, similar to other studies [41]. The time-domain analysis of RR intervals revealed no differences between that ME/CFS patients and healthy controls in any of the time-domain or frequency-domain parameters (Table 2). However, although the differences are not statistically significant, the control group shows higher values in all parameters than the ME/CFS group.

### 3.4. Correlation and Regression Analyses (Men)

Table 3 displays the correlation analysis between self-reported symptoms and HRV parameters for the total sample of men (*n* = 51). The only significant relationships are negative correlations between the COMPASS-31 gastrointestinal factor and SDNN (*p* = 0.042), RMSSD (*p* = 0.037), pNN50 (*p* = 0.04) and LF (*p* = 0.006), and between the physical scale of FIS-40 and pNN50 (*p* = 0.049).

A simple linear regression analysis revealed clear differences among the study participants, since ME/CFS patients showed significant relationships between the physical factor of FIS-40 and HRV parameters, such as SDNN (Figure 2A, β = −0.487, *p* = 0.0047), RMSSD (Figure 2B, β = −0.394, *p* = 0.0258), LF (Figure 2C, β = −0.537, *p* = 0.0015), HF (Figure 2D, β = −0.421, *p* = 0.0165), and pNN50 (β = −0.378, *p* = 0.033). No such associations were found for any HRV domain in healthy controls (Figure 2).

Table 4 shows Pearson correlations between global scores of self-reported PSQI, NCQ, FIS-40, COMPASS-31, and HADS questionnaires. All correlations were positive, highly significant, and higher than 0.73 (*p* < 0.001). Most correlation coefficients were higher than 0.8 (Table 4).

### 3.5. Differential Gender Effects and Interactions on Clinical Parameters

The data from men in the current study were combined with the data obtained from women in a previous study [14] into a unique data set and subsequently analyzed to evaluate significant gender effects and interactions with healthy or ME/CFS conditions.

Table 5 shows the baseline demographic and clinical parameters for men and women by group, and also shows significant gender effects and/or significant ‘gender*condition’ interactions (two-way ANOVA analysis). All variables, except age and BMI, showed a significant condition effect (Control vs. ME/CFS), with a level of significance of *p* < 0.01 for SAP and DAP and *p* < 0.001 for all other clinical variables. Considering only gender, without considering the group or experimental condition, women had lower SAP (*p* < 0.001), increased HR (*p* = 0.004), increased vasomotor domain score (*p* = 0.034) and a marginally significant increase in secretomotor domain score (*p* = 0.061) in the COMPASS-31, compared with men (Table 5). The only variable that showed a differential gender effect of the health condition was NCQ (*p* = 0.023), revealing that in women, ME/CFS patients showed a greater increase of neurovegetative complaints compared with the control group than the corresponding increase showed by men (see NCQ values in Table 5).

### 3.6. Gender Effects and Interactions on HRV

Table 6 shows the HRV parameters for men and women by group, and also shows significant gender effects and/or significant ‘gender*condition’ interactions (two-way ANOVA analysis).

In Figure 3 and Figure 4, we combine the HRV parameters for women [14] with the new ones obtained from men. Interestingly, the values of HF and its derivatives LF/HF and HFnu revealed significant gender effects (Table 6). Moreover, HF and HFnu presented significant ‘gender*condition’ interactions (*p* < 0.05), and LF/HF a marginally significant ‘gender*condition’ interaction (*p* = 0.065; Table 6). Post-hoc comparisons on LF revealed that the two groups of men did not show any difference, nor were differences found with control women or female ME/CFS patients (Figure 4A). By contrast, female ME/CFS patients had decreased HF and HFnu values compared to control women, whereas the two male groups did not differ between them or the female ME/CFS patients. The control group of women showed higher values on these parameters (Figure 4B,D). Post-hoc comparisons of the LF/HF ratio showed the same between-group differences, but in an inverse relationship, thus the female control group had the lowest ratio and differed from all the other groups (Figure 4C).

## 4. Discussion

This study was designed to explore HRV analysis as an objective, non-invasive and easy-to-apply marker of ME/CFS using mHealth technology. We analyzed the relationship of HRV parameters with self-reported symptoms of fatigue, autonomic dysfunction, subjective sleep quality, and anxiety/depression symptoms in men with ME/CFS. Also, we analyzed differential gender effects on healthy vs. ME/CFS conditions, comparing the new data on men from the current study with the data on women obtained in a previous study [14] combined into a unique data set. Finally, we briefly explore mHealth devices and compare them with what has been done in this article.

### 4.1. Analysis for Men

Our study is one of the few that has analyzed ME/CFS disorder in a reasonably large sample of men and compared it with a sample of women under experimental control conditions. Our results revealed that male ME/CFS patients self-reported severe fatigue, autonomic dysfunction, decreased sleep quality, and increased anxiety and depression symptoms, together with increased higher heart rate and blood pressure values at rest compared with male healthy controls. Specifically, the differences were more than 7 points in the NCQ score, more than 36 points in the COMPASS-31 total score, more than 120 points in the FIS-40 total score, around 10 points more in the PSQI total score, and 20 points more in the HADS total score. At the clinical level, all these differences indicate severe symptomatology in male ME/CFS patients, an aspect that is poorly documented in the literature. Besides, there was a high association between the male questionnaire scores themselves, indicating a close relationship between all the symptoms, consistent with the strong association that appeared in female ME/CFS patients [14].

We did not find any significant difference in HRV parameters in male ME/CFS patients compared with male healthy controls. However, all time-domain parameters (RRmean, SDNN, RMSSD, and pNN50) had a higher value in the control group, indicating greater cardiac variability (although significance was not achieved). It might be possible for them to reach significance in a larger sample, since they are consistent with the increased cardiac variability of control participants reported in previous studies [42]. Thus, the greater cardiac variability in the male control group would be related to better health, in agreement with other studies [43], since in resting conditions, a “healthier heart” shows greater cardiac variability and a lower level of heart rate, as it is related to a greater activation of the parasympathetic system and better physical condition. The values of the parameters analyzed went in the same direction as in our previous study comparing female ME/CFS patients vs. female healthy controls [14].

In the current study, the correlations between HRV parameters and self-reported fatigue scores were not as consistent as they were in the study in women. Notably, the only significant correlation that appeared between HRV parameters and self-reported symptoms in the male sample was between the physical scale of the FIS-40 and the gastrointestinal scale of the COMPASS-31. In the male sample, the FIS-40 physical scale was negatively associated with pNN50. Since this parameter indicates the percentage of normal successive RR intervals in which the change exceeds 50 ms, that negative relationship indicates that increased fatigue is associated with shorter variation between successive RR intervals, meaning less HRV. In addition, the gastrointestinal domain of autonomic dysfunction was negatively associated with SDNN, RMSSD, and pNN50 (the three main HRV indices of the time-domain analysis) and the LF component. It is considered that pNN50 and RMSSD are measures of parasympathetic activity, whereas SDNN reflects both sympathetic and parasympathetic modulation of heart rate. Regarding the LF (absolute power of the low frequency band) and HF (power of the high frequency band) components, vagal activity is the major contributor to the HF component, whereas the LF component is more controversial although considered by some as a marker of sympathetic activity; furthermore, diminished RMSSD and pNN50 can be interpreted as decreased vagal activity [44]. Overall, those relationships had the expected sign in our study: High fatigue or high gastrointestinal dysfunction were associated with less HRV variability, thus suggesting that those HRV variables are susceptible to be further explored as physiological biomarkers for fatigue symptoms in men.

Nonetheless, a closer inspection of the regression analysis suggested that physical fatigue could work as a biomarker in men. This analysis detected specific relationships for male ME/CFS patients that did not appear in the male healthy controls. Only the male ME/CFS patients showed a significant relationship between the physical scale of FIS-40 and HRV parameters. Inverse relationships with time-domain parameters, such as SDNN and RMSSD, line up with predicted symptomatology of higher physical fatigue with less cardiac variability, in agreement with other studies [7,43,45]. However, the results of the frequency-domain parameters are not so clear, since both HF and LF showed a similar inverse relationship with the physical fatigue score of FIS-40. The negative relationship between HF and physical fatigue in male ME/CFS patients is consistent with those of the temporal-domain, indicating more severe physical fatigue associated with less parasympathetic predominance, in line with previous works [14]. In apparent inconsistency, the same type of relationship appeared between LF and physical fatigue, when it would be expected that more physical fatigue should be related to more sympathetic predominance. However, both vagal and sympathetic activity can contribute to LF variability [39], and LF values could be influenced by respiratory sinus arrhythmia (RSA) [46]. Overall, the two parameters assessing parasympathetic and vagal activity, RMSSD, and HF, respectively, showed a robust relationship with self-reported physical fatigue severity in male ME/CFS patients, but not in healthy males. SDNN and LF, also showed the same pattern, but the interpretation of these two parameters is more controversial. In any case, such relationships had the expected sign, such that less HRV was associated with more severe fatigue symptoms. These results suggest that SDNN, RMSSD, LF, and HF could have certain predictive value for physical fatigue self-perception in male ME/CFS patients, RMSSD, and HF to a greater, and SDNN and LF to a lesser extent.

### 4.2. Analysis of Gender Differences

There are very few studies that have analyzed gender differences in ME/CFS patients. In the previously mentioned review on malfunctioning of the ANS [13], the only study in the review that examined gender differences found that healthy women had a different cardiovascular response to upright tilt compared to healthy men and that this could indicate a predisposition of women to orthostatic intolerance. However, these differences between male and female ME/CFS patients were not observed in another study. In their review about malfunctioning of the ANS in patients with ME/CFS (743 patients), Cauwenmergh et al. [13] accounted for a total of 74% of women in the samples from the reviewed studies. Only one study aimed to analyze gender differences; all the rest of the studies grouped men and women indiscriminately as patients.

In our study, to further investigate differential gender outcomes, we analyzed the two male and female samples in a two-way (2 × 2) design with a health condition (control, ME/CFS) and gender (male, female) as factors. Overall, women show lower SAP, increased HR, increased NCQ symptoms, and increased vasomotor domain score compared with men. Analyzing the clinical parameters, the only variable that showed a differential gender effect depending on health condition was NCQ. On average, female ME/CFS patients had an NCQ score of 9.71 points NCQ above the level of female healthy controls, whereas male ME/CFS patients scored 7.71 points above the level of male healthy controls. These results would indicate a higher level of autonomic dysfunction in female vs. male ME/CFS patients. Likewise, our results revealed gender differences for SAP, DAP, HR, and autonomic secretomotor and vasomotor dysfunctions between men and women. Women had lower systolic and diastolic blood pressures and higher heart rates compared with men.

Regarding HRV analysis, the profile of male ME/CFS patients differs from that of female ME/CFS patients, and robust differences were shown between female healthy controls and female ME/CFS patients, the latter group exhibiting lower values of time- and frequency-domain HRV parameters than the former [14]. The fact that these differences in HRV parameters did not reach significance in the study with men could be explained according to some studies reporting gender differences in short-term HRV [9,47]. Voss et al. [9] analyzed the influence of gender and age on short-term HRV in healthy subjects (782 women and 1124 men). They found significant modifications of the HRV indices, especially in the frequency domain, according to gender for people in an age range between 25 and 49 years, which would include those in our sample. For his part, Vaschillo et al. [47] also found gender differences related to HRV parameters when applying biofeedback techniques.

The correlations between HRV parameters and self-reported fatigue symptom scores that appeared in men are quantitatively consistent with those of women, since the sign of the correlations indicated less cardiac variability associated with more severe symptomatology as in the study on women. Thus, the relationship with time-domain HRV parameters (SDNN, RMSSD, and pNN50) showed the same trend in both genders. It can be interpreted in terms of greater severity of physical symptomatology is associated with worse cardiac variability, which agrees with other studies. To this effect, [42] consider that time-domain HRV parameters are biomarkers of psychophysiological and cardiovascular health; Mather and Thayer [8] proposed that the high amplitude oscillations in heart rate positively affect neural networks in the brain; and McCraty and Shaffer [48] found that higher levels of HRV are associated with health, adaptability, resilience, and self-regulatory capacity.

Regardless of health condition, all HF-related parameters revealed significant gender effects. The total sample of women showed higher HF and HFnu values than men, and lower LF/HF values (indicating that HF weighs more on women). This would indicate greater parasympathetic activation among women, in line with what other works expose [22,23,24]. However, when we analyzed gender differences considering the interaction with health status, we find specific differences for fatigued women, pointing out that this group is driving the differences. ME/CFS women showed lower HF and HFnu values than control women, whereas the two male groups did not differ between them nor from ME/CFS women (Figure 4B and Figure 2D). Thus, ME/CFS women appear to behave just like men by showing less activation of the parasympathetic system than healthy women. This lower parasympathetic predominance is also observed in the higher values of the LF / HF proportion for ME/CFS women, as well as for men (Figure 4C). Therefore, everything indicates that ME/CFS disorder has differential effects in women, decreasing the usual parasympathetic activation in healthy women to a level typical for men.

Regardless of gender, the total sample of ME/CFS patients showed significantly lower values of cardiac variability than controls. All time-domain HRV parameters (RRmean, SDNN, RMSSD, and pNN50) indicated a worse autonomic balance in ME/CFS patients than in healthy participants. Indeed, HRV is considered an index of autonomic resilience because it reflects the ability to recover from exposure to both physical and psychological stressors [7,43,45]. HRV has been demonstrated as a robust biomarker of psychophysiological health, good fitness level, and well-being [8,49,50,51]. In fact, high HRV is considered a good biomarker of physical health [42], or a good preventive indicator of physical injuries [52], whereas low HRV has been associated with, among other things, high risk of cardiovascular diseases [53], diabetes [54], cognitive impairment [55], and psychiatric disorders [56,57], such as depression or anxiety [58,59].

### 4.3. mHealth Technology for HRV Analysis

The diagnosis and monitoring of ME/CFS can be expensive and can take a lot of time and equipment (e.g., an exercise intolerance test with an ergometric test of maximum oxygen consumption, or a complicated neuropsychological battery for the study of cognitive impairment). Instead, based on the method and results of our study, we show that a 5-min HRV test at rest can provide useful information interpreting an objective, non-invasive and easy-to-apply marker of ME/CFS using mHealth technology. This tool can help in the diagnosis, monitoring, and clinical prediction of fatigue severity, especially in women with ME/CFS. More generally, it has already been shown that mHealth systems are useful to remotely detect changes in health parameters as HRV or heart rate (HR) in the context of the activity or daily living. HRV can evaluate symptom severity and progression, stability, and treatment responses [60]. For example, a recently published study analyzes the HRV for more than eight million people collected with a sensor based on photoplethysmography activity [61]. It shows how the HRV parameters (in the temporal and frequency domains) change with age, gender, and during the day. Another study analyzes the effect of adding HRV biofeedback to an app-based, remote intervention for depression and how it can enhance the treatment outcomes [62]. More studies have been published recently combining HRV and mHealth technologies [63,64,65,66]. However, one of the main limitations of those studies is the high sensitivity to the body movement and the presence of artifacts as that most of them rely on a photoplethysmography sensor for detecting the heartbeats or the inter-beat cardiac intervals [67,68]. The gold standard for HRV analysis is based on the lab-ECG signal. However, this recording method requires the use of electrodes and leads, so it is expensive, uncomfortable for the patient, and requires some training. As an alternative to the lab-ECG, they are recently appearing portable user devices based on mHealth technology, detecting ECG signals deriving the RR interval values for HRV analysis [38,68,69].

## 5. Conclusions

The total sample results, including women and men, robustly revealed consistent differences between ME/CFS patients and healthy controls in demographic and clinical parameters. ME/CFS patients felt severely fatigued, complained about all autonomic neurovegetative symptoms and dysfunctions, reported subjective poor sleep quality, and clinical depression and anxiety symptoms. Moreover, all HRV parameters showed a significant impairment in ME/CFS patients compared with healthy control participants. Time- and frequency-domain parameters behaved differently in terms of differential gender effects. Neither of the time-domain indices presented a significant gender effect and/or a significant interaction for gender vs. health condition. By contrast, frequency-domain parameters (HF, LF/HF, and HFnu) presented a robust significant gender effect, and HF and HFnu presented a significant interaction for gender vs. health condition. To sum up, HRV analysis is presented as a good way to analyze autonomic responses, and our results suggest that it can be useful for the diagnosis of ME/CFS. According to Cauwenmergh et al. [13], more sensitive and direct measurements of the autonomic activity may be necessary to establish whether slight autonomic dysfunction is involved in ME/CFS. HRV analysis meets this requirement, and according to our results, can help to diagnose in a simple way: (a) The severity of physical symptoms in ME/CFS patients of both genders based on time-domain HRV parameters (SDNN, RMSSD, pNN50); and (b) the level of parasympathetic autonomic dysfunction in ME/CFS women based on frequency-domain HRV parameters (HFnu, LF/HF). In conclusion, we show that HRV analysis using mHealth technology is an objective, non-invasive tool that can be useful for diagnosis, monitoring, and clinical prediction of fatigue severity, especially in women with ME/CFS. However, further studies are needed to corroborate whether such gender differences in HRV are associated with and predict fatigue severity and autonomic cardiac dysfunction in women with ME/CFS over time.

## Figures and Tables

**Figure 1 sensors-21-03746-f001:**
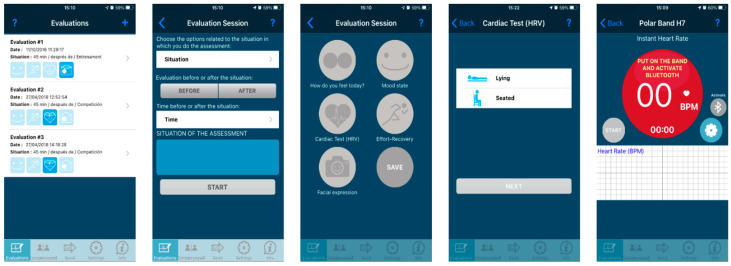
Screenshots sequence corresponding to the custom-made application (FitLab^®^ App). The first screen (left) shows all the stored recordings in the app pending to be synchronized with the server and the possibility to start a new one (“+” symbol). The next three screens ask for the situation where the recording takes place. The last screen (on the right) shows the instantaneous heart rate (in BPM) and the RR series.

**Figure 2 sensors-21-03746-f002:**
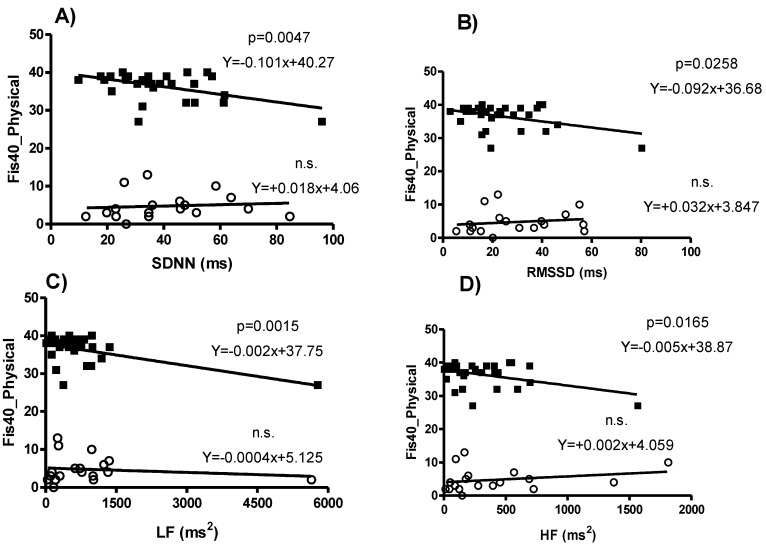
Simple regression analysis between physical fatigue perception (FIS-40) and HRV parameters differentiating ME/CFS patients (*n* = 32) and Control men (*n* = 19). Physical FIS-40 score is significantly explained from (**A**) SDNN (*p* = 0.005), (**B**) RMSSD (*p* = 0.026), (**C**) LF (*p* = 0.002), and (**D**) HF (*p* = 0.016), for ME/CFS patients (black squares, upper regression lines), but not for those healthy controls (white circles, bottom regression lines).

**Figure 3 sensors-21-03746-f003:**
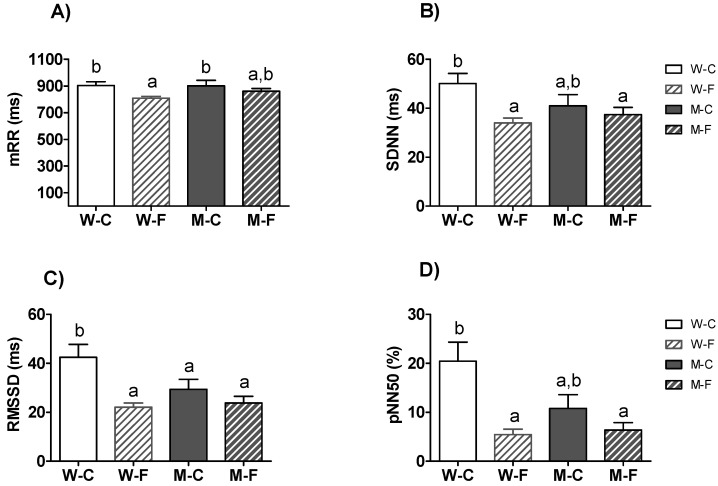
Comparison of the HRV time-domain indices in the whole sample. Mean ± SEM of (**A**) mean of RR intervals (meanRR), (**B**) standard deviation of all RR intervals (SDNN), (**C**) root mean square of differences of successive RR intervals (RMSSD), and (**D**) the proportion derived by dividing the number of interval differences of successive RR intervals greater than 50 ms by the total number of RR intervals (pNN50). a,b Mean values with unlike letters were significantly different between groups (two-way ANOVA and Duncan’s post hoc comparison, *p* < 0.05). W-C: Healthy control women (*n* = 25); W-F: ME/CFS women (*n* = 44); M-C: Healthy control men (*n* = 18); M-F: ME/CFS men (*n* = 32).

**Figure 4 sensors-21-03746-f004:**
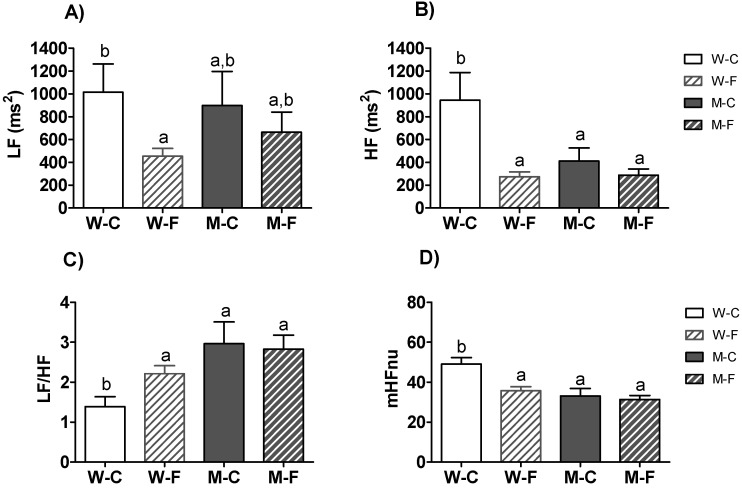
Comparison of the HRV frequency-domain indices in the sample. Mean ± SEM of (**A**) power of the low frequency band (LF), (**B**) power of the high frequency band (HF), (**C**) LF/HF ratio, and (**D**) normalized HF value (HFnu). a,b Mean values with unlike letters were significantly different between groups (two-way ANOVA and Duncan’s post hoc comparison, *p* < 0.05). W-C: Healthy control women (*n* = 25); W-F: ME/CFS women (*n* = 44); M-C: Healthy control men (*n* = 18); M-F: ME/CFS men (*n* = 32).

**Table 1 sensors-21-03746-t001:** Baseline demographic and clinical parameters among participants (men).

Variable	Controls (n = 19)	ME/CFS (n = 32)	*p*-Value
Age (years)	47.32 ± 1.51	47.38 ± 1.52	N.S.
BMI (kg/m^2^)	24.69 ± 0.80	23.69 ± 0.51	N.S.
SAP (mmHg)	122.1 ± 2.43	131.8 ± 2.54	0.014
DAP (mmHg)	77.91 ± 1.51	82.38 ± 1.68	0.054
HR (beats/min)	62.79 ± 1.33	70.13 ± 1.85	0.007
NCQ (number)	0.26 ± 0.10	7.97 ± 0.46	<0.001
**FIS-40**			
Global score (0–160)	11.68 ± 4.02	135.8 ± 3.91	<0.001
Physical	2.47 ± 1.05	36.50 ± 0.62	<0.001
Cognitive	3.58 ± 1.00	34.72 ± 0.72	<0.001
Psychosocial	5.63 ± 2.06	64.59 ± 2.41	<0.001
**COMPASS-31**			
Global score (0–100)	20.57 ± 2.93	56.83 ± 2.42	<0.001
Orthostatic intolerance	3.11 ± 0.41	7.56 ± 0.36	<0.001
Vasomotor	0 ± 0	1.28 ± 0.26	<0.001
Secretomotor	0.58 ± 0.21	3.94 ± 0.29	<0.001
Gastrointestinal	5.84 ± 1.03	11.0 ± 0.90	0.001
Bladder	0.58 ± 0.18	3.62 ± 0.48	<0.001
Pupillomotor	3.16 ± 0.70	9.69 ± 0.63	<0.001
**PSQI**			
Global score (0–21)	4.32 ± 0.67	14.28 ± 0.77	<0.001
Subjective sleep quality	0.53 ± 0.14	2.28 ± 0.14	<0.001
Sleep latency	0.53 ± 0.18	1.84 ± 0.18	<0.001
Sleep duration	0.95 ± 0.16	1.88 ± 0.19	0.001
Habitual sleep efficiency	0.42 ± 0.23	1.72 ± 0.22	<0.001
Sleep disturbances	1.00 ± 0.11	2.22 ± 0.11	<0.001
Sleeping medication	0.32 ± 0.13	1.91 ± 0.24	<0.001
Daytime dysfunction	0.58 ± 0.14	2.44 ± 0.14	<0.001
**HADS**			
Global score (0–42)	7.26 ± 1.0	27.38 ± 1.36	<0.001
Anxiety	5.21 ± 0.70	14.03 ± 0.67	<0.001
Depression	2.05 ± 0.49	13.34 ± 0.85	<0.001

Baseline self-reported outcome scores (global and subscales) of core symptoms, as explained in Methods 2. Values express mean ± SEM for each item. Abbreviations: BMI, Body mass index; SAP, systolic arterial pressure; DAP, diastolic arterial pressure; HR, heart rate; NCQ, neurovegetative complaints questionnaire, FIS-40, 40-item fatigue index scale; COMPASS-31, 31-item abbreviated composite autonomic symptom score; PSQI, Pittsburgh sleep quality index; HADS, Hospital Anxiety and Depression Scale.

**Table 2 sensors-21-03746-t002:** HRV parameters among participants (men).

Variable	Controls (n = 19)	ME/CFS (n = 32)	*p*-Value
RR mean (ms)	901.6 ± 41.0	861.3 ± 20.5	N.S.
SDNN (ms)	41.02 ± 4.52	37.38 ± 2.99	N.S.
RMSSD (ms)	29.37 ± 4.04	23.84 ± 2.67	N.S.
pNN50 (%)	10.79 ± 2.82	6.37 ± 1.52	N.S.
LF (ms^2^)	897.5 ± 298.8	663.6 ± 177.3	N.S.
HF (ms^2^)	411.1 ± 115.0	287.4 ± 54.8	N.S.
LF/HF	2.96 ± 0.55	2.83 ± 0.35	N.S.
HFnu	33.11 ± 3.73	31.36 ± 2.03	N.S.

Values are presented as means ± SEM. Abbreviations of HRV parameters: MeanRR, mean of RR intervals; SDNN, standard deviation of all RR intervals; RMSSD, root mean square of differences of successive RR intervals; pNN50, the proportion derived by dividing the number of interval differences of successive RR intervals greater than 50 ms by the total number of RR intervals; LF, low frequency band (0.04–0.15 Hz); HF, high frequency band (0.15–0.40 Hz); LF/HF ratio; HFnu, normalized HF value.

**Table 3 sensors-21-03746-t003:** Pearson correlation coefficients (r) between HRV parameters and self-reported measures (*n* = 51 men).

	Mean RR	SDNN	RMSSD	pNN50	LF	HF	LF/HF	HFnu
**PSQI**								
Sleep quality	−0.086	−0.084	−0.128	−0.121	−0.076	−0.108	−0.052	0.014
Sleep latency	−0.158	−0.076	−0.118	−0.113	−0.088	−0.055	−0.011	0.047
Sleep duration	−0.16	−0.166	−0.246	−0.19	−0.166	−0.256	0.037	−0.059
Habitual sleep efficiency	−0.119	−0.123	−0.146	−0.06	−0.152	−0.035	−0.066	0.115
Sleep disturbances	−0.12	−0.164	−0.188	−0.201	−0.209	−0.136	−0.092	0.076
Sleeping medication	−0.234	−0.072	−0.125	−0.092	−0.166	−0.051	0.009	0.04
Daytime dysfunction	−0.062	−0.081	−0.126	−0.176	−0.142	−0.095	−0.091	0.059
Global score	−0.175	−0.136	−0.193	−0.167	−0.181	−0.127	−0.046	0.056
**NCQ**	−0.232	−0.185	−0.235	−0.241	−0.152	−0.213	0.06	−0.126
**FIS-40**								
Physical	−0.173	−0.184	−0.239	−0.279*	−0.167	−0.223	−0.055	−0.018
Cognitive	−0.176	−0.146	−0.208	−0.258	−0.121	−0.193	−0.027	−0.026
Psychosocial	−0.201	−0.199	−0.237	−0.278	−0.135	−0.215	−0.048	−0.008
Global score	−0.189	−0.183	−0.232	−0.276	−0.141	−0.213	−0.045	−0.016
**HADS**								
Anxiety	−0.123	−0.101	−0.085	−0.086	−0.07	−0.064	−0.139	0.107
Depression	−0.228	−0.236	−0.231	−0.224	−0.134	−0.214	−0.003	−0.026
Global score	−0.189	−0.183	−0.173	−0.17	−0.11	−0.153	−0.066	0.035
**COMPASS-31**								
Orthostatic intolerance	−0.14	−0.094	−0.141	−0.124	−0.074	−0.079	−0.095	0.019
Vasomotor	−0.214	−0.079	−0.075	−0.121	0.073	−0.046	−0.008	0.016
Secretomotor	−0.143	−0.22	−0.253	−0.235	−0.223	−0.19	0.074	−0.08
Gastrointestinal	−0.061	−0.288 *	−0.296 *	−0.291 *	−0.382 **	−0.221	−0.193	0.195
Bladder	−0.107	−0.142	−0.178	−0.192	−0.126	−0.197	0.006	−0.078
Pupillomotor	−0.089	−0.122	−0.147	−0.18	−0.156	−0.118	−0.116	0.066
Global score	−0.148	−0.19	−0.23	−0.222	−0.194	−0.167	−0.085	0.032

Significance: * *p* < 0.05; ** *p* < 0.01.

**Table 4 sensors-21-03746-t004:** Pearson correlations between global questionnaire scores for the total sample (*n* = 51 men).

	NCQ	FIS-40	HADS	COMPASS-31
PSQI	0.788 **	0.819 **	0.817 **	0.825 **
NCQ		0.872 **	0.770 **	0.849 **
FIS40			0.871 **	0.817 **
HADS				0.737 **

** Significance (*p* < 0.001).

**Table 5 sensors-21-03746-t005:** Baseline demographic and clinical parameters (mean ± SEM) for men and women by group (data for women from Escorihuela et al., 2020 [14]).

	Male		Female		GENDER Dif.
Variable	Controls (n = 19)	ME/CFS (n = 32)	Controls (n = 25)	ME/CFS (n = 45)	*p*-Value
Age (years)	47.32 ± 1.51	47.38 ± 1.52	44.96 ± 1.30	46.41 ± 0.84	N.S.
BMI (kg/m^2^)	24.69 ± 0.80	23.69 ± 0.51	23.77 ± 0.61	24.59 ± 0.69	N.S.
SAP (mmHg)	122.1 ± 2.43	131.8 ± 2.54 *	115.2 ± 2.15	121.2 ± 1.99 *	<0.001 ^a^
DAP (mmHg)	77.91 ± 1.51	82.38 ± 1.68 *	74.45 ± 1.56	79.56 ± 1.38 *	0.055 ^a^
HR (beats/min)	62.79 ± 1.33	70.13 ± 1.85 *	67.71 ± 1.93	74.72 ± 1.21 *	0.004 ^a^
NCQ (num)	0.26 ± 0.10	7.97 ± 0.46 **	0.40 ± 0.15	10.11 ± 0.28 **	0.023 ^b^
**FIS-40**					
Global score (0–160)	11.68 ± 4.02	135.8 ± 3.91 **	17.12 ± 3.25	140.9 ± 1.79 **	<0.001
Physical	2.47 ± 1.05	36.50 ± 0.62 **	4.60 ± 0.94	36.95 ± 0.39 **	<0.001
Cognitive	3.58 ± 1.00	34.72 ± 0.72 **	4.48 ± 1.03	35.73 ± 0.66 **	<0.001
Psychosocial	5.63 ± 2.06	64.59 ± 2.41 **	8.04 ± 1.46	68.27 ± 1.04 **	<0.001
**COMPASS-31**					
Global score (0–100)	20.57 ± 2.93	56.83 ± 2.42 **	27.31 ± 2.42	80.10 ± 2.91 **	<0.001
Orthostatic intolerance	3.11 ± 0.41	7.56 ± 0.36 **	2.52 ± 0.25	7.45 ± 0.31 **	<0.001
Vasomotor	0 ± 0	1.28 ± 0.26 **	0.48 ± 0.21	1.93 ± 0.24 **	0.034 ^a^
Secretomotor	0.58 ± 0.21	3.94 ± 0.29 **	0.76 ± 0.18	4.73 ± 0.18 **	0.061 ^a^
Gastrointestinal	5.84 ± 1.03	11.0 ± 0.90 **	5.60 ± 0.74	13.45 ± 0.68 **	0.001
Bladder	0.58 ± 0.18	3.62 ± 0.48 **	0.32 ± 0.11	3.48 ± 0.32 **	<0.001
Pupillomotor	3.16 ± 0.70	9.69 ± 0.63 **	2.96 ± 0.46	10.32 ± 0.55 **	<0.001
**PSQI**					
Global score (0–21)	4.32 ± 0.67	14.28 ± 0.77 **	4.52 ± 0.63	15.05 ± 0.57 **	<0.001
Subjective sleep quality	0.53 ± 0.14	2.28 ± 0.14 **	0.56 ± 0.12	2.23 ± 0.14 **	<0.001
Sleep latency	0.53 ± 0.18	1.84 ± 0.18 **	0.72 ± 0.17	1.89 ± 0.16 **	<0.001
Sleep duration	0.95 ± 0.16	1.88 ± 0.19 *	0.92 ± 0.17	2.05 ± 0.13 **	0.001
Habitual sleep efficiency	0.42 ± 0.23	1.72 ± 0.22 **	0.56 ± 0.22	1.95 ± 0.17 **	<0.001
Sleep disturbances	1.00 ± 0.11	2.22 ± 0.11 **	1.04 ± 0.07	2.27 ± 0.13 **	<0.001
Sleeping medication	0.32 ± 0.13	1.91 ± 0.24 **	0.44 ± 0.12	2.55 ± 0.11 **	<0.001
Daytime dysfunction	0.58 ± 0.14	2.44 ± 0.14 **	0.44 ± 0.12	2.55 ± 0.11 **	<0.001
**HADS**					
Global score (0–42)	7.26 ± 1.0	27.38 ± 1.36 **	5.15 ± 0.70	26.68 ± 1.41 **	<0.001
Anxiety	5.21 ± 0.70	14.03 ± 0.67 **	3.96 ± 0.41	13.73 ± 0.73 **	<0.001
Depression	2.05 ± 0.49	13.34 ± 0.85 **	1.16 ± 0.29	12.95 ± 0.68 **	<0.001

* Differences with controls for the same gender (*p* < 0.05). ** Differences with controls for the same gender (*p* < 0.001). ^a^ Gender differences for all samples. ^b^ Gender differences depending on the group (gender x group significant interaction).

**Table 6 sensors-21-03746-t006:** HRV parameters for men and women by group (data for women from Escorihuela et al., 2020 [14]).

	Male		Female		*p*-Value (ANOVA)	
Variable	M-Controls (n = 19)	M-ME/CFS (n = 32)	W-Controls (n = 25)	W-ME/CFS (n = 45)	Gender	Gender by Group
RRmean (ms)	901.6 ± 41.0	861.3 ± 20.5	904.40 ± 27.63	809.40 ± 13.54 *	N.S.	N.S.
SDNN (ms)	41.02 ± 4.52	37.38 ± 2.99	50.06 ± 4.16	33.97 ± 2.03 **	N.S.	N.S.
RMSSD (ms)	29.37 ± 4.04	23.84 ± 2.67	42.49 ± 5.25	22.09 ± 1.72 **	N.S.	0.071
pNN50 (%)	10.79 ± 2.82	6.37 ± 1.52	20.46 ± 3.89	5.44 ± 1.09 **	N.S.	N.S.
LF (ms^2^)	897.5 ± 298.8	663.6 ± 177.3	1014.60 ± 247.5	453.10 ± 68.40 *	N.S.	N.S.
HF (ms^2^)	411.1 ± 115.0	287.4 ± 54.8	944.90 ± 241.8	274.73 ± 42.47 **	0.037	0.036
LF/HF	2.96 ± 0.55	2.83 ± 0.35	1.39 ± 0.25	2.21 ± 0.20 *	<0.001	0.065
HFnu	33.11 ± 3.73	31.36 ± 2.03	49.07 ± 3.31	35.77 ±1.99 *	N.S.	0.035

Values are shown as mean ± SEM. * Differences with controls for the same gender (*p* < 0.05). ** Differences with controls for the same gender (*p* < 0.001). “Gender”: Gender differences for all samples. “Gender by Group”: Gender differences depending on the group (gender x group significant interaction). Abbreviations of HRV parameters: MeanRR, mean of RR intervals; SDNN, standard deviation of all RR intervals; RMSSD, root mean square of differences of successive RR intervals; pNN50, the proportion derived by dividing the number of interval differences of successive RR intervals greater than 50 ms by the total number of RR intervals; LF, low frequency band (0.04–0.15 Hz); HF, high frequency band (0.15–0.40 Hz); LF/HF ratio; HFnu, normalized HF value.

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
