# Peer review of "Analysis of Gender Differences in HRV of Patients with Myalgic Encephalomyelitis/Chronic Fatigue Syndrome Using Mobile-Health Technology"

_sensors, 2021, doi:10.3390/s21113746_

Round 1

Reviewer 1 Report

This is a well-structured paper presenting the results of the heart rate variability study on a large number of patients (121 persons). The results have been treated correctly both from the point of view of measurement procedures and from the point of view of subsequent signal processing, so that the results obtained present a good starting point for the differentiation of the diagnosis between men and women in diseases such as chronic fatigue.  However, a number of questions arise from its review and should be explained in the article.

  1. From the point of view of the measurement procedure, has the same test been repeated to different users, at different times of the day or on different days to ensure that the variability of the measurements in each user is statistically lower than the variability found per group?
  2. Concerning signal processing: Are there five minutes of artifact-free (gold-standard) cardiac signal from all users? (gold-standard) Can these possible artifacts affect the quality of the estimation of heart rate variability?
  3. If the results are validated, how could they be used in the diagnosis and monitoring of the diseases presented in the article? Has it been tested whether the heart rate changes as the disease progresses? Could there be problems with the fact that the heart rate may be influenced by other factors/diseases?

Reviewer 2 Report

This paper presents the cohort research results that show the relation between ME/CFS and self-reported symptoms combined with HRV parameters. The cohort survey is well organized, and the conclusions are clearly presented. The following revisions should be considered for publication.

1. (Lines 95, 511, and 563) On Line 563, it concludes that the parasympathetic dysfunction can be diagnosed for women with ME/CFS using the relevant HRV parameters, HFnu and LF/HF. This is reasonable because women have high reactivity of the parasympathetic nervous system than men according to Line 95. On Line 511, it describes that RMSSD and HF relevant to the parasympathetic activity has a robust relation with fatigue severity for men with ME/CFS patient. According to Line 95, however, men have high reactivity of the sympathetic nervous system rather than the parasympathetic one. Such high reactivity of the sympathetic nervous system is not consistent with the relation between fatigue severity and the parasympathetic activity for men. Please explain a reason for this inconsistency.

2. (Line 338) It describes that only one variable shows the gender effect on the health condition. That is NCQ. According to Table 5, however, high p-value is observed also for the vasomotor and the secretomotor. Like NCQ, the vasomotor and the secretomotor show the gender effect, in which each variable shows for women a higher increase from the control to the ME/CFS groups than the same variables for men. Please explain exclusion of the vasomotor and secretomotor variables from the variables showing the gender effect.

3. (Line 400) The Discussion section is too long to clearly understand the survey results. Please separate the whole section into several sub-sections for easy understanding. Especially, one sub-section should be allocated to the explanation on the gender effect because it is one of main topics in this paper.

4. (Line 453) Please explain shortly why better health causes the greater cardiac variability for men.

5. I see the following typos in the manuscript.

1) (Line 420) Our results revealed than => Our results revealed that

2) (Line 488) men?. => men.

3) (Line 528) Fig 2B and 2c => Figs 3(b) and (d)

4) (Line 531) Fig 2D => Fig 3(c)

Reviewer 3 Report

In general, I think it is very good and interesting research.

I have some minor concerns about the paper.

1.- I think the relationship between chronic fatigue symptoms and heart rate variability needs to be adjusted for confounders, specifically age.
2.- The authors did simple linear regression analysis. Why was not a multiple linear regression analysis done?
3.- the authors used several scales and questionnaires. However, they do not indicate if they are validated in the Spanish language of Spain, if they used the original version, if they used the validated Spanish version, or another alternative.

On the other hand, I have some major concerns about the paper.

4.- The article is presented in a very "medical" way. Not much is explained about the sensor (cardiac chest band) or the program built on the iOS platform, which I think could have been relevant for this journal. For this journal, the article has to improve on the above.

5.- I think a related work section is missing. It is necessary to review other mHealth devices and compare them with what has been done in this article. Furthermore, neither in the results nor in the discussion sections there are concrete references to other mHealth devices to perform the same or something similar to what was done in this study.

Round 2

Reviewer 1 Report

The article can be accepted in its present form. The changes introduced by the authors respond correctly to the points addressed in the review.

Reviewer 3 Report

No more comments.